# Overexpression of miR-192 Inhibits In Vitro Porcine Embryo Development by Inducing Oxidative Stress Damage and Impairing Mitochondrial Function

**DOI:** 10.3390/ani15010046

**Published:** 2024-12-27

**Authors:** Fan He, Mingguo Li, Fan Chen, Rong Zhou, Mengfan Qi, Binbin Fu, Huapeng Zhang, Qingchun Li, Yanzhen Bi, Tao Huang

**Affiliations:** 1College of Animal Science and Technology, Shihezi University, Shihezi 832000, China; 18299075532@163.com (F.H.); 16699230598@163.com (M.L.); qimengfan1998@sina.com (M.Q.); fbb15546093387@sina.com (B.F.); 13699358979@163.com (H.Z.); 18899598025@163.com (Q.L.); 2Hubei Key Laboratory of Animal Embryo Engineering and Molecular Breeding, Institute of Animal Husbandry and Veterinary, Hubei Academy of Agricultural Sciences, Wuhan 430070, China; fanchen@hbaas.com; 3The State Key Laboratory of Animal Biotech Breeding, Institute of Animal Science, Chinese Academy of Agricultural Sciences, Beijing 100193, China; rongzhou@caas.cn; 4Xinjiang Pig Breeding Engineering Technology Research Center, Xinjiang Tecon Husbandry S&T Co., Ltd., Changji 831100, China

**Keywords:** pig, miR-192, early embryo development

## Abstract

This study investigates the role of miR-192 in early embryo development of pig. The aim was to understand how altering miR-192 levels impacts embryonic growth and development. The researchers injected miR-192 agomir (a molecule that increases miR-192 activity) into pig embryos activated without fertilization and observed its effects. The results showed that elevated miR-192 levels hindered the embryos’ ability to progress through critical stages, such as the 2-cell, 4-cell, and blastocyst stages. This impairment was linked to cellular stress, mitochondrial dysfunction, and an increase in reactive oxygen species (ROS), causing significant damage to the embryos. Furthermore, the overexpression of miR-192 suppressed the expression of key genes involved in maintaining pluripotency and developmental potential, such as YY1 and NANOG. These findings suggest that miR-192 has a detrimental effect on early pig embryo development, offering new insights into its role in reproduction. This research may have implications for improving livestock breeding and understanding the molecular mechanisms underlying early embryonic development.

## 1. Introduction

Mammalian early embryonic development is a complex, multi-tiered regulatory process, with miRNAs and functional genes forming an extensive regulatory network widely involved in zygotic genome activation (ZGA), cleavage, and cell fate determination [1,2]. miRNAs are a class of short non-coding RNAs, typically 17–25 nucleotides in length, which regulate gene expression by targeting the 3′ untranslated regions and/or coding regions of messenger RNAs (mRNAs) [3,4]. miR-192, a highly conserved miRNA across species, exhibits elevated expression levels in extracellular vesicles (EVs) derived from uterine lumen fluid (ULF) of pregnant sows on days 9 and 13 of gestation [5,6]. Notably, the EVs in pig ULF primarily originate from uterine epithelial cells and are internalized by the porcine embryos [5]. Previous studies have also confirmed that the expression level of miR-192 in serum-derived EVs from pregnant sows on day 15 is significantly higher than that in non-pregnant sows [7]. In humans, the concentration of miR-192-5p in the culture medium of blastocysts following intracytoplasmic sperm injection (ICSI) is markedly higher than in the medium from non-blastocyst embryos [8]. These findings suggest that miR-192 may act as a key communication molecule between the mother and fetus through EV cargo, playing a role in regulating early embryonic development and the implantation process.

miR-192/miR-192-5p is also expressed in pre-implantation embryos across multiple species [9]. Through screening various datasets, we observed distinct expression patterns of miR-192/miR-192-5p across different stages of early embryonic development in humans, mice, and pigs. In human embryos (zygote, 4-cell, and 8-cell), miR-192-5p follows an up–down–up expression pattern, while in mouse embryos (2-cell, 4-cell, and 8-cell), the expression increases progressively. In porcine embryos (oocyte, 4/8-cell, and blastocyst), miR-192 expression shows slight fluctuations, but no significant differences are observed between stages [10,11,12,13]. Although the endogenous expression of miR-192 in porcine embryos does not exhibit significant temporal changes, exogenous miR-192, such as that derived from ULF-derived EVs, may be internalized by the embryos and play a specific functional role.

Our previous research has demonstrated that YY1 is one of the key target genes of miR-192 [14]. YY1 is a transcription factor with dual functions, acting both as an activator and repressor of gene expression and plays a crucial role in regulating the pluripotency of early embryonic cells [15]. Studies have shown that YY1 regulates nucleosome organization at enhancer-like regions and mediates dynamic nucleosome remodeling during early mouse development. The absence of YY1 leads to developmental delays during the transition from morula to blastocyst. Furthermore, although YY1 does not affect the mRNA levels of pluripotency factors such as OCT4 and SOX2 in mouse embryonic stem cells, YY1 is essential for the localization and stability of OCT4 and SOX2 proteins, which is a prerequisite for the specification of the inner cell mass (ICM) lineage [16].

In this study, we hypothesize that the role of miR-192 in embryonic development is not only related to regulating endogenous expression but also involves the communication mechanism between the mother and the fetus. Therefore, we employed a miR-192 overexpression strategy (agomir and agomir NC) to simulate the maternal transfer of miR-192 in order to explore its potential impact on embryonic development. Our results indicate that miR-192 agomir impairs mitochondrial function in 4-cell stage parthenogenetic pig embryos, resulting in abnormal levels of mitochondrial membrane potential and ROS levels, as well as inducing embryo cell apoptosis, leading to a significant decrease in embryo cleavage and blastocyst rates. Additionally, our findings suggest that miR-192 agomir downregulates the mRNA expression levels of YY1 and the pluripotency regulator NANOG in embryos, which may be a key factor in the impaired developmental capacity of pig embryos caused by miR-192 overexpression.

## 2. Materials and Methods

### 2.1. Porcine Oocyte Collection and In Vitro Maturation

Porcine ovaries were obtained from a local abattoir in Shihezi and promptly transported to the laboratory in saline solution at a temperature range of 37–39 °C. All procedures involving animals were approved by the Animal Care Committee of Shihezi University, China (A2019-144-01).

Follicular fluid was prepared from antral follicles measuring 3–6 mm in diameter using a 10 mL syringe equipped with a 1.2 mm diameter needle. Cumulus oocyte complexes (COCs) with homogeneously granulated cytoplasm and multiple layers of cumulus cells were selected for in vitro maturation (IVM). In brief, COCs were washed three times in HEPES and then transferred to a four-well culture plate containing the IVM medium, with approximately 30 COCs per well. The plate was incubated for 44 h at 38.5 °C in a 5% CO_2_ incubator to facilitate oocyte maturation. The IVM medium was composed of a TCM-199 medium (M4530, Sigma, St. Louis, MO, USA) supplemented with 10% porcine follicular fluid, 5 µg/mL insulin–transferrin–selenium, 10 ng/mL epidermal growth factor, 0.6 mM L-cysteine, 0.2 mM pyruvate, 25 µg/mL kanamycin, and 5 IU/mL each of equine chorionic gonadotropin (eCG) and human chorionic gonadotropin (hCG).

### 2.2. Parthenogenetic Activation (PA) and In Vitro Culture

Oocytes were treated with 0.2% hyaluronidase for 3 min to remove cumulus cells, select oocytes with polar bodies, and transfer them into an activation medium composed of 0.3 M D-mannitol, 0.175 mM CaCl_2_·2H_2_O, and 0.05 mM MgCl_2_·6H_2_O. PA was induced by applying two direct current pulses of 1.2 kV/cm for 30 milliseconds each to the oocytes. Following electroactivation, the oocytes were incubated in a PZM-3 medium containing 7.5 µg/mL cytochalasin B for 3 h to inhibit the release of the second polar body. Subsequently, the oocytes were washed, approximately 30 oocytes were placed in each well of a 4-well plate, and 500 µL of a PZM-3 medium containing 0.4% bovine serum albumin (BSA) was added. The oocytes were then cultured in an incubator at 38.5 °C with 5% CO_2_.

### 2.3. Microinjection and Embryo Development Assessment

Six hours post-electroactivation, microinjection was performed on the oocytes. Using a microinjection apparatus (Eppendorf, Hamburg, Germany), miR-192 agomir and a scrambled negative control (20 pmol/μL, Ribobio, Guangzhou, China) were injected into the activated oocytes. According to the previous study, the success of the injection was confirmed by observing the cytoplasmic movement induced by the procedure [3,17,18]. Thereafter, the oocytes were returned to continue cultivation, with the day of oocyte activation designated as Day 0. Cleavage rates were assessed on Day 1 (24 h) and Day 2 (48 h), followed by an evaluation of blastocyst formation on Day 7 (168 h).

### 2.4. Measurement of ROS

In accordance with the manufacturer’s guidelines for the ROS assay kit (S0033S, Beyotime, Shanghai, China), dichlorodihydrofluorescein diacetate (DCFH-DA) was prepared by diluting it in a serum-free culture medium at a 1:1000 ratio, ensuring thorough mixing with a vortex. Subsequently, embryos at the 4-cell stage were carefully transferred to the staining solution. The embryos were incubated at 37 °C for 20 min within a cell culture incubator. Post-staining, the embryos underwent three successive rinses with the serum-free culture medium for 2 min each to remove excess dye. Once washed, the embryos were placed in a four-well plate filled with the serum-free culture medium in preparation for imaging. The fluorescence emitted by the embryos was examined, and their images were captured utilizing a fluorescence microscope. The quantification of fluorescence intensity was performed using ImageJ software Version 1.54 (National Institute of Health, Bethesda, MD, USA), providing an objective measurement of ROS levels within the embryos.

### 2.5. Mitochondrial Membrane Potential Assessment

The mitochondrial membrane potential (MMP) of 4-cell stage embryos was evaluated using a dedicated assay kit (C2006, Beyotime, Shanghai, China). The procedure commenced with the incubation of embryos in a working concentration of JC-1 dye at 38 °C for 30 min. Post-incubation, the embryos were rinsed using a serum-free culture medium to eliminate unbound dye. Utilizing an inverted fluorescence microscope (Nikon Eclipse TE2000-U, Tokyo, Japan) coupled with a digital camera, images were captured that depicted both red and green fluorescence signals. The red fluorescence, signifying JC-1 aggregates and indicative of active mitochondria, was juxtaposed with the green fluorescence, which is associated with JC-1 monomers and reflects inactive mitochondrial states. The ImageJ software was then applied to quantify the ratio of these fluorescence intensities, providing a measure of mitochondrial membrane potential.

### 2.6. AnnexinV-FITC Staining

Early-stage apoptosis in porcine blastocysts was assessed using the Annexin V-FITC Apoptosis Detection Kit (C1062S, Beyotime, Shanghai, China). The methodology involved washing 20 to 30 blastocysts with PBS containing 0.01% PVA (*w*/*v*), followed by incubation in 100 μL of binding buffer supplemented with 5 μL of Annexin V-FITC for 15 min at room temperature. After thorough rinsing, the blastocysts were immediately examined under an inverted fluorescence microscope for signs of apoptosis.

### 2.7. Prediction of miR-192 Target Genes and Functional Analysis

The target genes of miR-192 were predicted using the TargetScan (v7.2) and miRanda (v3.3a) databases. Predicted target genes were filtered by selecting those with a TargetScan score ≥ 60 and a miRanda energy score ≤ −17. The overlap between the results from both algorithms was used to ensure high-confidence predictions.

Subsequently, Gene Ontology (GO) and Kyoto Encyclopedia of Genes and Genomes (KEGG) enrichment analyses were performed to explore the biological functions and pathways associated with the identified target genes. GO enrichment analysis categorized the genes into biological processes, molecular functions, and cellular components, while KEGG pathway analysis identified the relevant signaling pathways. Both analyses were conducted using the OmicStudio tool [https://www.omicstudio.cn/tool (accessed on 10 December 2024)], with the top 20 enriched terms visualized.

For protein–protein interaction (PPI) analysis, a PPI network was constructed using the STRING database (version 12.0) with a confidence score threshold of ≥ 0.70. K-means clustering was applied to extract core clusters, and the network was visualized using Cytoscape (version 3.9.0). The degree centrality method was used to identify hub genes, which were subsequently ranked based on their centrality within the network.

### 2.8. Quantitative RT-PCR

Total RNA was extracted from a sample of 20 embryos at the 4-cell stage or 10 blastocysts using the RNAprep Pure Micro Kit (4992859, Tiangen, Beijing, China). For miRNA reverse transcription, the miRcute Plus miRNA First-Strand cDNA Kit (4992909, Tiangen, Beijing, China) was employed. mRNA reverse transcription was carried out using the HiScript II Q RT SuperMix for qPCR (R223-01, Vazyme, Nanjing, China). Subsequent qRT-PCR analysis was performed on the LightCycler^®^ 96 system (Roche, Basel, Switzerland). In the miRNA quantification, each 20 μL reaction consisted of 10 μL of 2 × miRcute Plus miRNA PreMix (FP411, Tiangen, Beijing, China), 0.4 μL of each primer, and 1 μL of cDNA; U6 was used as the internal reference gene. For mRNA quantification, each 20 μL reaction included 10 μL of 2 × ChamQ Universal SYBR qPCR Master Mix (Q711, Vazyme, Nanjing, China), 0.4 μL of each primer, and 1 μL of cDNA; RPLP0 was used as the internal reference gene. Relative expression levels were determined using the 2^−ΔΔCT^ method. The primers used for the qRT-PCR reactions are listed in Table 1.

### 2.9. Statistical Analysis

Each experiment was independently replicated a minimum of three times to ensure the reliability of the results. SPSS 20.0 software was used for the independent sample *t*-test to evaluate the differences between groups in cleavage rate, blastula rate, relative gene expression level, and relative fluorescence intensity, and GraphPad Prism 8 software was used for plotting. A *p*-value of less than 0.05 was considered statistically significant.

## 3. Results

### 3.1. miR-192 Inhibits the Developmental Capacity of Porcine Parthenogenetic Embryo

We initiated our study by investigating the impact of miR-192 on embryonic development through the microinjection of miR-192 agomir and a non-targeting agomir control (agomir NC) into porcine parthenogenetic embryos that had been activated. Quantification of miR-192 expression was conducted 48 h post-injection, with cleavage rates documented at 24 and 48 h and blastocyst formation evaluated at 168 h. Our findings revealed a significantly elevated expression of miR-192 in embryos treated with the miR-192 agomir compared to the agomir NC group (Figure 1a, *p* < 0.01). Furthermore, the miR-192 agomir treatment group significantly reduced the developmental progression of porcine parthenogenetic embryos to the 2-cell, 4-cell, and blastocyst stages (Table 2 and Figure 1b,c, *p* < 0.01).

### 3.2. miR-192 Induces Embryonic Oxidative Stress and Increases ROS Levels

ROS levels are known to markedly influence the early stages of embryogenesis. An overabundance of ROS can result in oxidative harm to essential biomolecules, including lipids, proteins, and DNA. To determine if miR-192 triggers an increase in ROS, we utilized DCFH-DA to measure ROS formation in 4-cell stage embryos treated with miR-192 agomir. The findings indicated elevated average ROS fluorescence intensity in the miR-192 agomir-treated embryos compared to the control group (Figure 2a,b). Additionally, to assess the broader effects of miR-192 on oxidative stress, we evaluated the mRNA levels of key antioxidant enzymes, specifically CAT and SOD1. A decrease in the mRNA levels of these enzymes was observed in embryos treated with miR-192 agomir (Figure 2c, *p* < 0.05). These findings indicate that miR-192 leads to oxidative stress in early embryos due to decreased expression of antioxidant enzymes and accumulation of ROS.

### 3.3. miR-192 Reduces Mitochondrial Membrane Potential

Mitochondrial membrane potential (MMP), a crucial biomarker for mitochondrial functionality, indicates apoptosis initiation when diminished. To evaluate the influence of miR-192 on mitochondrial function at the 4-cell embryonic stage, we employed JC-1 staining. This approach measures mitochondrial health by quantifying the fluorescence intensity ratio of JC-1 aggregates (indicating functional mitochondria with red fluorescence) to JC-1 monomers (suggesting mitochondrial depolarization with green fluorescence). Our findings indicate a pronounced decrease in membrane potential in embryos subjected to miR-192 agomir treatment relative to the control group (Figure 3a,b), which suggests impaired mitochondrial function.

### 3.4. miR-192 Promotes Apoptosis in Porcine Parthenogenetic Embryos

The accumulation of ROS and the decline in MMP suggest that miR-192 may induce embryo apoptosis. To elucidate this hypothesis, we employed Annexin-V-FITC staining and quantitative analysis to assess the apoptotic effects of miR-192 agomir on blastocyst-stage embryos. The results revealed that blastocysts in the miR-192 agomir group exhibited deeper Annexin-V staining (Figure 4a,b), indicating that miR-192 agomir induces apoptosis in blastocysts. Similarly, fluorescence quantitative assays supported this finding, showing an increase in BAX mRNA expression and a decrease in BCL-2 mRNA levels in blastocysts treated with miR-192 agomir, along with a significant increase in the BAX/BCL-2 ratio (Figure 4c). Therefore, the upregulation of miR-192 induces apoptosis in blastocysts, which is detrimental to early porcine embryo development.

### 3.5. miR-192 Suppresses the Expression of YY1 and Pluripotency in Pig Blastocysts

To investigate the impact of miR-192 on genes and pathways during porcine embryonic development, we predicted 290 high-confidence potential target genes of miR-192 (including YY1, a target gene identified in previous research) using the OmicStudio tool [https://www.omicstudio.cn/tool (accessed on 10 December 2024)] and performed GO and KEGG enrichment analyses. The results revealed that these target genes were significantly enriched in biological processes and molecular functions such as the regulation of pluripotency in stem cells (ko04550), the FoxO signaling pathway (ko04068), positive regulation of pyruvate dehydrogenase activity (GO:1904184), and ATPase binding (GO:0051117), suggesting that miR-192 may play a crucial role in maintaining pluripotency and regulating energy metabolism during embryonic development (Figure 5a–c and Appendix A).

Moreover, we constructed a high-confidence (confidence score ≥ 0.70) protein–protein interaction (PPI) network between the potential target genes of miR-192 and genes related to embryonic development using the STRING database (Version 12.0). Subsequently, K-means clustering was applied to extract clusters centered around H3-3B and YY1, which are closely associated with embryonic development, further highlighting their pivotal roles in the regulatory network of embryonic development (Figure 5d and Appendix A). To confirm whether miR-192 affects embryonic pluripotency, we evaluated its impact on the expression levels of YY1 and pluripotency marker genes SOX2, NANOG, and OCT4. qRT-PCR analysis indicated that miR-192 agomir significantly decreased the mRNA expression levels of YY1 and NANOG in porcine blastocysts while having no significant effect on OCT4 and SOX2 mRNA levels (Figure 5e,f, *p* < 0.05). Taken together, these findings provide evidence that miR-192 agomir may impair pluripotency in porcine parthenogenetic embryos by downregulating the expression of the downstream target YY1.

## 4. Discussion

Recent evidence indicates that miRNAs, as regulatory factors in zygote and early embryo reprogramming, play a crucial role in determining embryonic cell fate [19,20]. It has been reported that maternal-derived extracellular vesicles (EVs) in the early pregnancy uterine fluid carry high levels of miR-192, which can be effectively taken up by porcine embryos [5,6]. Small RNA expression profiles of early embryos in humans, mice, and pigs indicate that miR-192 is expressed to varying degrees throughout the developmental stages from the zygote to the blastocyst [10,11,12,13]. These data highlight that, in addition to the well-established role of embryo-derived miR-192, maternal miR-192 may also exert a regulatory function during embryonic development. Therefore, in this study, we simulated the maternal transfer of miR-192 by microinjecting miR-192 agomir into porcine parthenogenetic embryos, aiming to investigate the role of miR-192 in pre-implantation embryos.

In this study, miR-192 agomir injection significantly increased the expression level of miR-192 in embryos, resulting in a significant decrease in the cleavage rate and blastocyst formation rate of porcine parthenogenetic embryos, indicating that high expression of miR-192 impairs the developmental potential of porcine parthenogenetic embryos. In published studies, highly expressed miRNAs such as miR-143, miR-210, and miR-155 have been shown to inhibit to varying degrees the ability of porcine parthenogenetic embryos or fertilized eggs to develop into blastocysts, indicating that precise miRNA dosage control is a key factor in ensuring normal development of early pig embryos [3,18,21]. Dysregulated miRNA expression often has adverse effects on the embryonic development process. Although the overexpression of miR-192 has a negative effect on the development of porcine parthenogenetic embryos, a comprehensive analysis of the role and mechanism of miR-192 in early pig embryo development is still needed.

The depolarization of mitochondrial membrane potential disrupts the transfer of electrons to oxygen acceptors and leads to excessive ROS production [22]. As the center of energy metabolism, mitochondria generate a large amount of ATP through oxidative phosphorylation to support the development of pre-implantation embryos. Mitochondrial membrane potential is often used as a crucial indicator of mitochondrial function because the electrochemical gradient across the mitochondrial inner membrane is the basis for driving ATP synthesis [23]. In this study, we observed a significant depolarization of mitochondrial membrane potential in the miR-192 agomir group, severely impairing the ability of mitochondria to produce ATP, resulting in a substantial accumulation of reactive oxygen species (ROS). The mRNA expression of antioxidant factors CAT and SOD1 was also significantly suppressed, and the imbalance between ROS levels and the antioxidant system damaged mitochondrial function.

Additionally, the depolarization of mitochondrial membrane potential is considered an early event in cell apoptosis [22]. Therefore, we evaluated the impact of miR-192 on embryo apoptosis at the blastocyst stage, and the results of Annexin-V-FITC staining showed that the overexpression of miR-192 exacerbated the apoptosis signal. Previous studies have indicated that the membrane potential of mitochondria decreases due to the opening of the permeability transition pores (PTPs). These PTPs are formed by the binding of pro-apoptotic factor BAX with the adenine nucleotide translocator (ANT) in the inner membrane, causing changes in membrane permeability and depolarization, which release apoptotic factors (cytochrome c) and initiate a series of caspase-related pathways [24]. BCL-2 has been shown to inhibit apoptotic factors such as cytochrome c and mitochondrial apoptosis-inducing factors, suppressing programmed cell death [25]. In our study, miR-192 agomir led to an increase in BAX mRNA expression and suppression of BCL-2 mRNA expression in blastocysts, with a significantly elevated BAX/BCL-2 ratio, indicating that miR-192 may deepen the depolarization level of mitochondrial membrane potential by upregulating BAX and downregulating BCL-2, thereby altering mitochondrial inner membrane permeability, releasing apoptotic factors, and triggering the initiation of embryo apoptosis in pig blastocysts. In summary, our results demonstrate that miR-192 causes abnormal mitochondrial function in pre-implantation pig embryos, leading to a decrease in mitochondrial membrane potential and an increase in apoptosis in pig blastocysts.

YY1 is an indispensable multifunctional regulatory factor in mammalian embryonic development, maintaining early embryo development through multilayer epigenetic crosstalk [26]. In previous studies, YY1 has been identified as a downstream target gene of miR-192 [14,27,28]. In this study, the mRNA expression level of YY1 in early pig embryos was similarly downregulated by miR-192. Studies have shown that YY1 promotes enhancer-promoter interaction by selectively binding to low-methylated DNA sequences, participating in the regulation of chromatin remodeling and maintenance of pluripotency-related epigenetic landscapes in early mouse embryos. The depletion of YY1 leads to delayed or lethal development from morula to blastocyst in mice, greatly reducing birth rates [29,30]. Although it is not clear whether YY1 is involved in regulating chromatin remodeling in early pig embryos, our results suggest that miR-192 agomir may impair the developmental potential of early pig embryos by downregulating YY1. Additionally, YY1 mediates the interaction of pluripotency factors OCT4 and SOX2 at the post-transcriptional level. Although YY1 deficiency does not affect the mRNA levels of OCT4 and SOX2 in mouse embryonic stem cells, the localization and stability of OCT4 and SOX2 proteins require YY1 [16]. Similarly, our results also showed that while miR-192 agomir led to the downregulation of YY1 in pig blastocysts, there was no significant change in the mRNA expression of OCT4 and SOX2. However, the expression of another pluripotency factor, NANOG mRNA, was inhibited. Nevertheless, whether there are clear changes in the expression of OCT4 and SOX2 proteins was not addressed in this study and requires further investigation. These results suggest the possibility that miR-192 impairs the developmental potential and pluripotency of early pig embryos by downregulating its target gene, YY1.

## 5. Conclusions

Our results indicate that the overexpression of miR-192 has a negative impact on early pig embryo development. It can increase ROS accumulation, impair mitochondrial function, promote embryo apoptosis, and downregulate the expression of its target gene YY1 and pluripotency gene NANOG, thus impeding embryo development. In conclusion, this study reveals that the overexpression of miR-192 adversely affects early pig embryo development, providing new evidence for understanding the role of miR-192 in reproduction.

## Figures and Tables

**Figure 1 animals-15-00046-f001:**
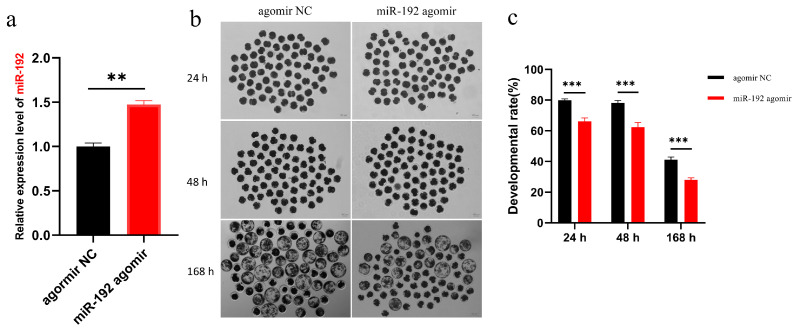
The impact of miR-192 on the developmental capacity of porcine parthenogenetic embryos. (**a**) qRT-PCR analysis depicts the expression levels of miR-192 in porcine parthenogenetic embryos treated with agomir NC and miR-192 agomir 48 h post-treatment. (**b**) Representative images of porcine parthenogenetic embryos treated with agomir NC and miR-192 agomir after 24, 48, and 168 h of in vitro culture. Scale bar = 100 μm. (**c**) The cleavage rate at 24 and 48 h and the blastocyst formation rate at 168 h were significantly reduced following treatment with miR-192 agomir in porcine parthenogenetic embryos. ** *p* < 0.01; *** *p* < 0.001. Results are presented as the mean ± SEM of at least three independent experiments.

**Figure 2 animals-15-00046-f002:**
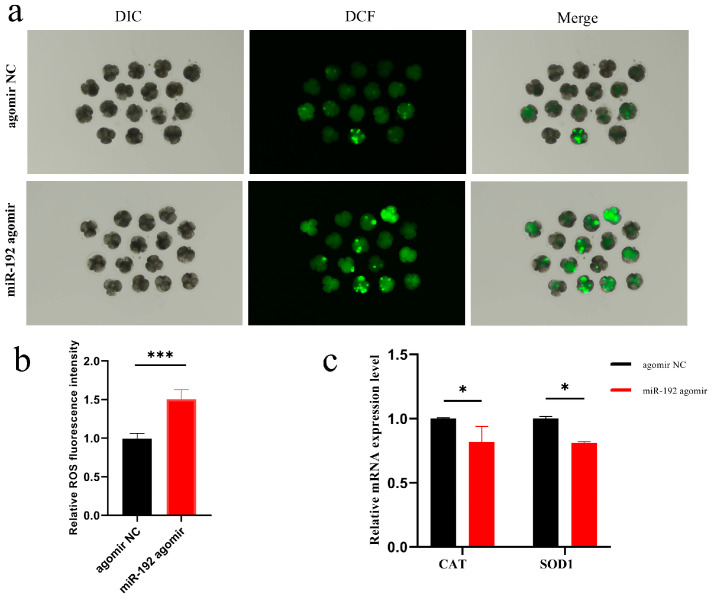
The effects of miR-192 on embryonic ROS levels and antioxidant enzyme expression. (**a**) The ROS fluorescence intensity in early embryos at the 4-cell stage (magnified 100 ×, scale bar = 100 µm). (**b**) The relative ROS levels in embryos at the 4-cell stage. (**c**) The relative mRNA expression levels of the antioxidant enzymes CAT and SOD1 in embryos at the 4-cell stage. * *p* < 0.05; *** *p* < 0.001. Results are presented as the mean ± SEM of at least three independent experiments.

**Figure 3 animals-15-00046-f003:**
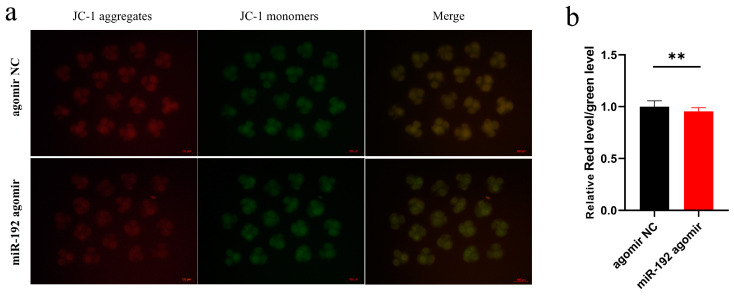
Impact of miR-192 on embryonic mitochondrial membrane potential. (**a**) Staining of 4-cell stage embryos with JC-1 aggregates (red fluorescence) and JC-1 monomers (green fluorescence) (magnification 100×, scale bar = 100 µm). (**b**) The fluorescence intensity ratio of JC-1 aggregates (red fluorescence) to JC-1 monomers (green fluorescence). ** *p* < 0.01. Results are presented as the mean values ± SEM of at least three independent experiments.

**Figure 4 animals-15-00046-f004:**
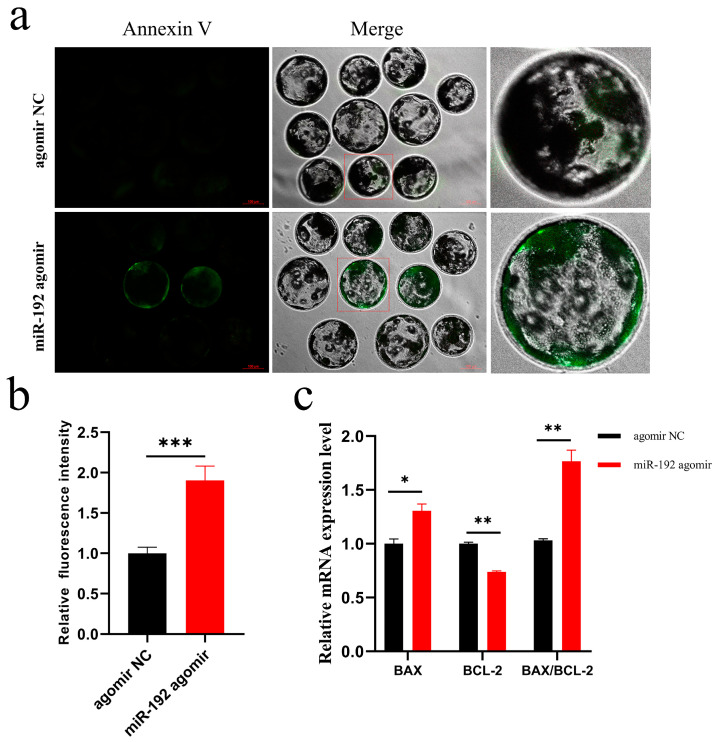
Impact of miR-192 on embryonic apoptosis. (**a**) Early apoptosis fluorescence signal in pig blastocyst-stage cells (magnification 100 ×, scale bar = 100 µm). (**b**) Relative level of Annexin-V-FITC fluorescence intensity. (**c**) Relative mRNA expression levels of BAX, BCL-2, and BAX/BCL-2 in blastocysts. * *p* < 0.05; ** *p* < 0.01; *** *p* < 0.001. Results are presented as the mean ± SEM of at least three independent experiments.

**Figure 5 animals-15-00046-f005:**
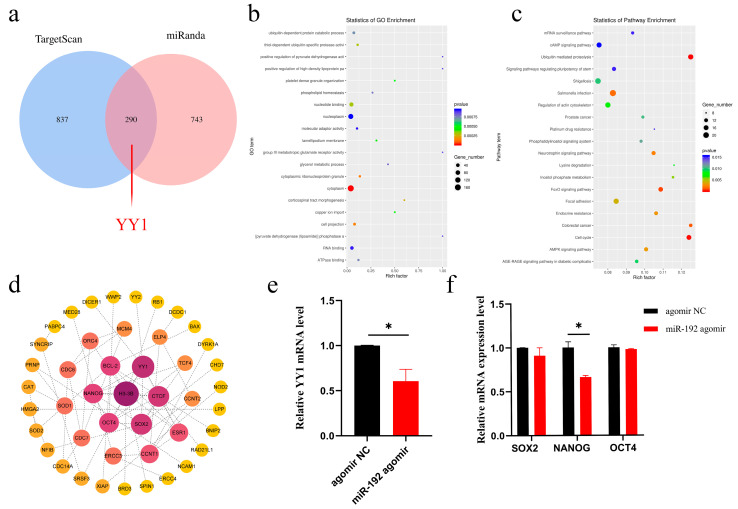
The effect of miR-192 on YY1 and pluripotency genes in the blastocyst stage. (**a**) miR-192 target genes were predicted using the TargetScan and miRanda databases, and the intersection was taken (TargetScan_score ≥ 60 and miranda_Energy ≤ −17). (**b**) The top 20 terms by gene ontology (GO) analysis. (**c**) The top 20 pathways by Kyoto Encyclopedia of Genes and Genomes (KEGG) pathway analysis. (**d**) The PPI network between miR-192 target genes and embryonic development-related genes was constructed using the STRING database and visualized using Cytoscape. Minimum required interaction score: high confidence (0.700). (**e**) Significant downregulation of YY1 mRNA expression by miR-192. (**f**) Relative mRNA expression levels of pluripotency-related genes. * *p* < 0.05. Results are presented as the mean ± SEM of at least three independent experiments.

**Table 1 animals-15-00046-t001:** Primer sequences used for real-time PCR.

Gene	Accession	Primer Sequences (5′ to 3′)	Product Size (bp)
miR-192	NR_038549.1	F:CCCTGACCTATGAATTGACAGCC	
U6	JN617885.1	F:CGCTTCGGCAGCACATATACTA	
		R:ATGGAACGCTTCACGAATTTGC	
CAT	XM_021081498.1	F:AACTGTCCCTTCCGTGCTA	202
		R:CCTGGGTGACATTATCTTCG	
SOD1	NM_214127.2	F:TTCCATGTCCATCAGTTTGG	107
		R:TGCCTCTCTTGATCCTTTGG	
BAX	XM_013977773.2	F:CGGGACACGGAGGAGGTTT	189
		R: CGAGTCGTATCGTCGGTTG	
BCL-2	XM_021077298.1	F:CAGGGACAGCGTATCAGAGC	156
		R:TTGCGATCCGACTCACCAAT	
YY1	XM_021099699.1	F:GAAGTGGGAGCAGAAGCAGGTG	149
		R:CGGAATAATCAGGAGGCGAGTTCTC	
SOX2	NM_001123197.1	F:CGCAGACCTACATGAACG	103
		R:TCGGACTTGACCACTGAG	
NANOG	XM_021092390.1	F:AGGACAGCCCTGATTCTTCCACAA	198
		R:AAAGTTCTTGCATCTGCTGGAGGC	
OCT4	XM_021097869.1	F:AAGCAGTGACTATTCGCAAC	136
		R:CAGGGTGGTGAAGTGAGG	
RPLP0	NM_001129964.2	F:GCTAAGGTGCTCGGTTCTTC	112
		R:GTGCGGACCAATGCTAGG	

**Table 2 animals-15-00046-t002:** Evaluation of development in embryos microinjected with miR-192.

Group	Number of Embryos	Number of Embryos Developed (Mean ± SEM, %)		
		2-Cell (24 h)	4-Cell (48 h)	Blastocyst (168 h)
agomir NC	210	167 (79.99 ± 1.16) ^A^	164 (78.09 ± 2.43) ^A^	86 (41.18 ± 2.40) ^A^
miR-192 agomir	206	136 (66.19 ± 1.78) ^B^	128 (62.38 ± 2.40) ^B^	58 (27.94 ± 1.20) ^B^

Results are presented as the mean ± SEM of at least three independent experiments. Differing uppercase letters (A, B) in the same column indicate a significant difference (*p* < 0.001).

## Data Availability

Data are contained within the article and Appendix A.

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
