# Peer review of "Overexpression of miR-192 Inhibits In Vitro Porcine Embryo Development by Inducing Oxidative Stress Damage and Impairing Mitochondrial Function"

_animals, 2024, doi:10.3390/ani15010046_

Round 1
Reviewer 1 Report
Comments and Suggestions for Authors
This research explores the impact of miR-192 on early porcine embryo development, with a particular focus on its role in oxidative stress, mitochondrial dysfunction, embryo apoptosis, and the regulation of pluripotency-related genes during embryo development. The study further clarifies that the functions of miR-192 in the early stages of sow pregnancy vary at different stages, providing new insights into the theoretical basis for the future application of miR-192 in animal reproduction and embryo engineering. These findings warrant further in-depth research and exploration. The following comments may be beneficial to optimize the article.
Comments:
1. Lines 47-48: If the "represents" here is replaced by "is" would be more direct and concise.
2. Lines 48-50: "results in the formation of" can be simplified as "forms" to improve conciseness, and "yielding" replaced with "eventually resulting in" to improve fluency.
3. Lines 68-70: It is recommended to adjust the logical order slightly to strengthen the focus of the information.
4. In the "Introduction" section, repetitive expressions can be streamlined. For example, the role of miR-192 is described multiple times in different paragraphs, and some repetition can be avoided to ensure that each paragraph adds new information.
5. Lines 94-95: "YY1 deficiency results in developmental delay during the transition from morula to blastocyst." It is suggested to add a further explanation or supplement to briefly explain the role of YY1 or the specific consequences of its absence to help readers better understand its importance.
6. Please correct all superscript and subscript issues in the "Materials and Methods" section.
7. Ensure consistency in units, terminology, and formatting. For example, "medium" is typically used in the singular form, and "oocyte" should be consistently used throughout the text.
8. The full term should be written out with the abbreviation in parentheses when first mentioned, for example, "parthenogenetic activation (PA)."
9. Lines 204: "Markedly elevated" can be simplified to "significantly higher," which is more precise and commonly used in scientific writing.
10. Lines 207-210: The sentence is long, so breaking it up and standardizing the presentation of statistical data (spacing between values and symbols) enhances readability.
11. The first paragraph of the discussion section provides an overview of the extensive background of existing research on miR-192. While this information is helpful, some sections can be lengthy, especially the detailed listing of the literature. In this section, we can appropriately simplify and retain the key and most relevant literature content to enhance the focus of information.
12. The main experimental results are summarized in the conclusion part, but the practical significance and application prospect of these results are not fully described. The discussion on the potential applications of miR-192 to regulate pig embryo development, such as its potential application in animal reproduction technology, or its implications in gene editing, cloning and other fields, can be considered.
Comments on the Quality of English LanguageThe authors are suggested to revise the manuscript more carefully to prohibit grammatical errors.
Reviewer 2 Report
Comments and Suggestions for Authors
In this manuscript, He and colleagues investigate the role of the microRNA Mir-192 in porcine preimplantation development. They do so by microinjecting an agomir of Mir-192 into parthenogentically activated pig oocytes. This led to an increase in levels of Mir-192 RNA, and decreased rate of in vitro development to blastocyst stage. They also observe increased levels of oxidative stress, and apoptosis markers. Expression levels of YY1, a known Mir-192 target was diminished and Nanog but not Sox2 or Oct4 mRNA levels were also decreased in the blastocyst.
Overall, this is a well-designed study that shows a detrimental role for overexpression of Mir-192 in porcine early embryonic development. The presentation of the results is clear and the conclusions are generally well-supported by the data. I just have a few recommendations detailed below:
· It would be useful to determine the expression level of Mir-192 in normal porcine preimplantation development. Is it highly or lowly expressed? Does it show dynamic changes across different stages?
· It is not clear to me why the authors used an agomir strategy to study the effect of too high levels of Mir-192, rather than attempting to deplete Mir-192 to address the role of endogenous Mir-192. It would be helpful if the authors could provide this rationale.
· On this note, I would suggest the title be changed to reflect the fact that too much or overexpression of Mir-192 inhibits embryonic development, rather than Mir-192 itself inhibits embryonic development, as Mir-192 is expressed in normal embryos (Fig. 1a), as far as I understand.
· Please could the authors clarify whether statistical testing is based on comparison of independent experiments? For example, in Figure 1c does the statistical test compare the means/medians of independent experiments or the total number of embryos? Why was the paired T test employed? Similarly, are standard error bars presented in the Figures based on the number of independent experiments or the number of individual embryos? The same concerns apply to the other Figures.
· The authors describe using the 2-ΔΔCT method for quantitative RT-PCR. Which primer sets were used for normalization (refers to Figures 1a, 2c, 4c, 5a/b)?
· In the methods both sections 2.7 and 2.8 are described as ‘Quantitative RT-PCR’
Reviewer 3 Report
Comments and Suggestions for Authors
This study provides a valuable contribution to understanding the molecular mechanisms governing embryogenesis. The study also connects miR-192 with ROS accumulation, mitochondrial dysfunction, apoptosis, and reduced pluripotency via YY1 downregulation. This multi-faceted approach strengthens the conclusions.
This study clearly depicts the role of MiR-192 in embryo development and provides important insights into miRNA-mediated regulation of early embryogenesis. In addition, providing more detailed assessments of mitochondrial morphology or dynamics under miR-192 overexpression conditions could offer additional insights into mitochondrial impairment.
Although the research study is designed I suggest the authors to explore other potential target genes or pathways regulated by miR-192, as it might interact with a network of genes affecting embryo development using bioinformatics approaches.
Reviewer 4 Report
Comments and Suggestions for Authors
Simple summary: no comments
Abstract: well written, no changes necessary
Introduction: Line 68 needs a reference, so did line 76-78
Material and methods: I don’t found the number of oocytes used for this study. Approximate number at least.
In section 2.7, authors should mention their reference/housekeeping gene used in qPCR.
Results: Lines 207-211 can be a small table
Discussion: no comments
Conclusion: since pig is an important livestock animal, authors may add one or two lines about how this work impact in broad scenario.
Round 2
Reviewer 2 Report
Comments and Suggestions for Authors
All my concerns have now been addressed and I am happy to recommend publication of this manuscript